## [Decision Letter · Decision Letter 0]

5 Jun 2025

PGENETICS-D-25-00207

Plagl1 regulates the retinal progenitor cell to Müller glial cell transition

PLOS Genetics

Dear Dr. Schuurmans,

Thank you for submitting your manuscript to PLOS Genetics. After careful consideration, we feel that it has merit but does not fully meet PLOS Genetics's publication criteria as it currently stands. Therefore, we invite you to submit a revised version of the manuscript that addresses the points raised during the review process.

Please submit your revised manuscript within 60 days Aug 04 2025 11:59PM. If you will need more time than this to complete your revisions, please reply to this message or contact the journal office at plosgenetics@plos.org. Please include the following items when submitting your revised manuscript:

We look forward to receiving your revised manuscript.

Kind regards,

Fengwei Yu

Section Editor

PLOS Genetics

Fengwei Yu

Section Editor

PLOS Genetics

Aimée Dudley

Editor-in-Chief

PLOS Genetics

Anne Goriely

Editor-in-Chief

PLOS Genetics

**Journal Requirements:**

At this stage, the following Authors/Authors require contributions: Yacine touahri, Alissa Pak, Luke David, Joseph Hanna, Hedy Liu, Yucheng Xiao, Yaroslav Ilnytskyy, Edwin van Oosten, Nobuhiko Tachibana, Lata Adnani, Jiayi Zhao, Mary Hoffman, Rajiv Dixit, Dawn Zinyk, Cynthia Guidos, Volker Enzmann, Pengpeng Bi, Isabelle Aubert, Laurent Journot, Igor Kovalchuk, Yves Sauvé, Jeff Biernaskie, Chao Wang, Satoshi Okawa, and Antonio del Sol. Please ensure that the full contributions of each author are acknowledged in the "Add/Edit/Remove Authors" section of our submission form.

The list of CRediT author contributions may be found here: https://journals.plos.org/plosgenetics/s/authorship#loc-author-contributions

3) We noticed that you used the phrase 'data not shown' in the manuscript. We do not allow these references, as the PLOS data access policy requires that all data be either published with the manuscript or made available in a publicly accessible database. Please amend the supplementary material to include the referenced data or remove the references.

4) We do not publish any copyright or trademark symbols that usually accompany proprietary names, eg ©,  ®, or TM  (e.g. next to drug or reagent names). Therefore please remove all instances of trademark/copyright symbols throughout the text, including:

- ® on pages: 30, 31, 33, 34, and 37

- TM on pages: 30, 31, and 37.

5) Please upload all main figures as separate Figure files in .tif or .eps format. For more information about how to convert and format your figure files please see our guidelines:

Potential Copyright Issues:

i) Figures 2E, 7A, 7C, 7G, and 7H. Please confirm whether you drew the images / clip-art within the figure panels by hand. If you did not draw the images, please provide (a) a link to the source of the images or icons and their license / terms of use; or (b) written permission from the copyright holder to publish the images or icons under our CC BY 4.0 license. Alternatively, you may replace the images with open source alternatives. See these open source resources you may use to replace images / clip-art:

7) Please provide a detailed Financial Disclosure statement. This is published with the article. It must therefore be completed in full sentences and contain the exact wording you wish to be published.

1) Please clarify all sources of financial support for your study. List the grants, grant numbers, and organizations that funded your study, including funding received from your institution. Please note that suppliers of material support, including research materials, should be recognized in the Acknowledgements section rather than in the Financial Disclosure

2) State the initials, alongside each funding source, of each author to receive each grant. For example: "This work was supported by the National Institutes of Health (####### to AM; ###### to CJ) and the National Science Foundation (###### to AM)."

3) State what role the funders took in the study. If the funders had no role in your study, please state: "The funders had no role in study design, data collection and analysis, decision to publish, or preparation of the manuscript."

4) If any authors received a salary from any of your funders, please state which authors and which funders..

5)  Please ensure that the funders and grant numbers match between the Financial Disclosure field and the Funding Information tab in your submission form. Note that the funders must be provided in the same order in both places as well.

8) Please send a completed 'Competing Interests' statement, including any COIs declared by your co-authors. If you have no competing interests to declare, please state "The authors have declared that no competing interests exist". Otherwise please declare all competing interests beginning with the statement "I have read the journal's policy and the authors of this manuscript have the following competing interests:"

**Reviewers' comments:**

Reviewer's Responses to Questions

**Comments to the Authors:**

Reviewer #1: Review is uploaded as an attachment

Reviewer #2: The manuscript titled, “Plagl1 regulates the retinal progenitor cell to Müller glial cell transition” by Touahri, et al. claims to provide evidence that the transcription factor PLAGL1 regulates the transition of retinal progenitor cells to a Müller glial identity. The authors present a plethora of data, but this reviewer feels much of it is overinterpreted and several essential controls and confirmation experiments are missing.

SPECIFIC COMMENTS.

Figure. 1

Why did the authors not used PLAGL1 immunofluorescence (IF)? Their 2007 paper (Ma, et al.) used IF and they reported PLAGL1 protein expression in P7 and P21 MGs, RGCs, ACs, and HCs. In this current manuscript they highlight Plagl1 mRNA expression as being specific to MGs and astrocytes, but this is contradictory with their own previous IF finding.

The violin plots of the Blackshaw data are difficult to visualize and the change in Plagl1 mRNA seems subtle at best. Is there a better way to represent these data?

The authors use Sox9 mRNA in situ to make the claim that MNU-induced damage leads to ectopic Sox9 mRNA in the ONL that they interpret as injury-induced nuclear migration of glial cells into the ONL. These data do not support this conclusion. It’s possible that the change in Sox9 transcript localization is simply due to MG hypertrophy that occurs during MG gliosis and the MG processes extending into the ONL become more elaborate and contain Sox9 mRNA. Only IF for SOX9 and LHX2 can definitively make the conclusion that ectopic MGs are present in the ONL of the MNU damages retinas.

The authors claim that the increase in Plagl1 mRNA in response to MNU peaks at 6-12 hrs post-damage and then reduces from 24-72 hrs. which they claim is the peak of MG proliferation. I don’t agree with this interpretation. Firstly, the reference they cite analyzed rat retinas and it shows a large number or BrdU+ and pH3+ cells that express S100 and SOX9, respectively that are much above the levels that are seen in mouse MGs at 24-48 hrs. post-damage. I wonder if many of these cells are actually astrocyte and microglia rather than MGs. It is important for the authors to independently place their MNU injured mouse retina Plagl1 mRNA data in the context of a mouse MG proliferation timeline. This is a significant matter as the authors are trying to build a case that Plagl1 might normally inhibit MG proliferation.

Figure. 2.

In this figure, the authors claim that the mutant retinas contain proliferative MGs past the normal period during retinogenesis and that these cells eventually become MGs, amacrine cells, ganglion cells and rods. I do not feel the data fully support these claims.

BrdU alone is not a valid readout for cell proliferation as it can be taken up by cells that are dying or undergoing DNA repair. Ki67 and Caspase 3 immunofluorescence along with TUNEL labeling should be performed. Also, the mutant retinas are very dysplastic, and this could cause the infiltration of microglia and other immune cells which would also be proliferative. Furthermore, the BrdU+/Sox9+ cells could be astrocytes. The lamination defects appear quite severe by P7 suggesting the origin is much earlier and that the histological changes seen postnatally are secondary to that. Also, the co-localization of BrdU with Sox9, Pax6, and Rhodopsin is not convincing.

Figure 3.

The authors state, “These morphological abnormalities in Plagl1+/-pat retinas phenocopied various mutant mice with underlying Müller glia defects [21, 56, 58].” However, they do not cite the actual papers demonstrating this, but rather 3 review articles. It’s worth noting that their phenotype is quite similar to the retinal progenitor cells specific conditional knockout of Beta-catenin which suffer from defects in cell adhesion during development. It’s curious that the ERG defects seem subtle considering the architectural changes that are described.

Figure 4.

The authors state that, “upregulated Müller glia-enriched genes in P7 Plagl1+/-pat retinas included glial determinants and injury response genes of the Nfi [18], Sox [66], Lhx [68], Tead [69] and Vsx2 [70] TF families (Fig 4F,G). Notch pathway gene (Notch1,Notch2, Hes1, Hes5, Dll1, Jag2) expression was also elevated (Fig 4F,G), a signaling pathway that stimulates mammalian Müller glia to proliferate [71, 72] and is required for the initial proliferative response of mammalian Müller glia to injury [73].” I have 2 problems with this interpretation – 1. These gene are also expressed in retinal progenitor cells suggesting that the P7 mutant retinas could simply have persistent RPCs. 2. If an injury response is upregulated, why are the genes encoding ribosome protein subunits down? MGs responding to damage become hypertrophic and this leads to massive upregulation of ribosome protein expression. It’s also confusing why – if the MGs are proliferating – they downregulate translation as proliferative cells must increase biomass.

The FUNCAT experiment is not well-described. Why was the reference for the zebrafish system included versus the mouse system? What Cre was used to express the MetRS*? No validation experiments are shown (ex. + or - AHA). Fluorescence intensity is not a valid quantitative measure. The authors could easily perform (BONCAT) using biotin Western blots as a more quantitative readout. This method should be coupled with a more standard method of assessing translation such as puromycin labeling.

Without. retinal ChIP-seq, it’s impossible to know what’s primary versus secondary effects of Plagl1 loss.

The authors profiled mutant retinas at P7 when they claim MGs are proliferating, but the uncovered cell cycle DEGs do not shed much insight. For example, the mutant retinas show a reduction in Cyclin E2 and Cyclin G1. These factors would be expected to be upregulated in proliferative cells.

Figure 6.

It is very difficult to definitely make the claim that the mutant MGs express rod transcripts. It is well-known from sc-seq experiments of FAC sorted MGs, that rod transcripts often contaminate the samples and have to be computationally eliminated. Additional evidence such as cytometry for MG and rod markers would make these data more convincing.

Figure 7.

The MG CKOs do not make a convincing case at the cells undergo interkinetic nuclear migration (IKNM). During development, IKNK is coupled to the cell cycle. However, the CKO MG are described as non-proliferative. It is possible that disruption of the SOX9+ MG monolayer is due to overall disruption in retinal lamination and the MG are passively displaced out of the INL. Such an interpretation would fit with a general cell adhesion defect. It is also curious that the authors do not show Cre reporter expression in the CKOs. If displaced MGs were due to a primary phenotype impacting IKNM, the authors could provide additional evidence of this by assessing GFP expression. The expectation would be that the displaced MGs would be GFP+ (CKOs) and the GFP- MGs (that did not express Cre due to mosaicism) would be normally localized to the INL. If GFP- MGs were also displaced, this would argue for a secondary phenotype that non-cell autonomous.

I feel the DAPT explant culture is misinterpreted. The retinas were treated at P0 – a time when many progenitors are still present. The DAPT treatment is simply causing premature cell cycle exit and precocious neurogenesis at the expense of MGs. These data do not support the conclusion that Notch is required for the maintenance of MG fate.

Why did the authors choose to activate ICN with a GFAP-Cre AAV versus Glast-CreERT2? Again, it is not clear whether the disorganized MGs are secondarily due to changes in retinal lamination versus a primary defect in the MGs. Also, the AAVs cannot be considered 100% MG specific. In fact, the authors show mCherry expression in the ONL. Ectopic expression of ICN in neurons might cause cell death leading to changes in retinal lamination. In fact, at no point in this manuscript is cell death addressed. None of the phenotyping results are valid without assessment of cell death.

Finally, a Plagl1 gain-of-function experiment (either with AAV or electroporation) would be very informative in terms of assessing whether its activity is instructive for the MG fate.

**Have all data underlying the figures and results presented in the manuscript been provided?**

Reviewer #1: Yes

Reviewer #2: Yes

PLOS authors have the option to publish the peer review history of their article (what does this mean? ). If published, this will include your full peer review and any attached files.

**Do you want your identity to be public for this peer review?** For information about this choice, including consent withdrawal, please see our Privacy Policy .

Reviewer #1: No

Reviewer #2: No

**Figure resubmission:**
---

## [Decision Letter · Decision Letter 1]

15 Dec 2025

PGENETICS-D-25-00207R1

Plagl1 regulates the retinal progenitor cell to Müller glial cell transition

PLOS Genetics

Dear Dr. Schuurmans,

Thank you for submitting your manuscript to PLOS Genetics. After careful consideration, we feel that it has merit but does not fully meet PLOS Genetics's publication criteria as it currently stands. Therefore, we invite you to submit a revised version of the manuscript that addresses the points raised during the review process.

Please submit your revised manuscript within by Jan 14 2026 11:59PM. If you will need more time than this to complete your revisions, please reply to this message or contact the journal office at plosgenetics@plos.org. Please include the following items when submitting your revised manuscript:

We look forward to receiving your revised manuscript.

Kind regards,

Fengwei Yu

Section Editor

PLOS Genetics

Fengwei Yu

Section Editor

PLOS Genetics

Aimée Dudley

Editor-in-Chief

PLOS Genetics

Anne Goriely

Editor-in-Chief

PLOS Genetics

**Reviewers' comments:**

Reviewer's Responses to Questions

Reviewer #1: The Authors have done a good job providing additional data and restructuring the manuscript to minimize the over-interpretations of the data as presented. Most of the remaining comments are considered minor and reflect mechanisms to improve the dialog of the manuscript or statistical rigor of the data.

Most of the PLAGL1 and Sox9 up-regulation after MNU treatment in Figure 1G is potentially within the outer segments of photoreceptors or in extra-retinal tissue. Can the authors please comment on the specificity of the altered staining and how this relates to RNA transcript levels changed in sorted Müller glia? The results provided suggest that the up-regulation of Sox9 and Plagl1 is outside of Müller cells.

Figure 2C-D – Hard to determine co-localization of cells expressing BrdU (nuclear) and Rho (non-nuclear). Also, statistical comparisons of between percentages of BrdU+ cells co-expressing different marker genes is not a canonical analysis. Rho+ cells should also be Otx2+.

Figure 3K and M – displaying the normalized expression of pERK and Glul in a violin plot is unusual. Given that the number of replicates is less than or equal to 6, all replicate points should be plotted with standard deviation or standard error.

Can the authors postulate in the discussion as to why the A- and B-waves implicit timing is affected Plagl1 mutant retinas? The Authors suggest that the effect is Müller glia-specific. Please rationalize why this ERG phenotype would be consistent with Müller glia dysfunction.

Lines 355-364 – Authors have not established that Müller glia are proliferating.

Line 391 – Nfix and Vsx2 are not listed in Table4 as DARs and DEGs. They should be removed from this portion of the text.

Line 454-455 – shouldn’t it be 100% reduction in the mutants since the maternal allele is silenced due to imprinting? However, it is noted that the transcript is still produced even in the Bulk RNA-seq analysis

Line 465-470, 481-486 – Please use appropriate statistics to examine the significance of the effect. There are 8 samples (5 control and 3 mutants), please perform a t-test on the percentage of cells assigned to each cluster comparing WT and mutant. This will help determine if the effect is technical (due to one sample being over-represented in cluster 23 in mutants) or truly biological. Please do the same for the precursor populations (cluster 16) and for assessing the extent of proliferation (lines 481-486).

Line 492 - "endothelial cells also proliferate ectopically" – this is not discussed in the context of the results and the figures are not referenced (S7 fig) talking about these results.

The PCA analyses seem to represent the variance due to biological sex (Cs6-8) and read-depth (cs4-8). The authors should consider these parameters as alternate interpretations of the results.

Reviewer #2: I commend the authors for responding to most of my concerns. The manuscript is much improved. However, the authors should address lingering issues regarding Figure 2. Here, they show that the Plag1+/-pat retinas have persistent BrdU+/Sox9+ cells at P7, which they interpret as proliferating MG or delayed RPCs. They go on to support this interpretation by labeling P7 retinas with Ki67 and showing that the mutant retinas contain Ki67+ cells, whereas the control retinas do not (Figure 2E). However, the image they show is difficult to interpret. A few Ki67+ cells appear to be at the retinal periphery, and the labels for the ONL and INL are incorrect. It’s also unclear whether the WT image is of a comparable region. These images must be replaced. Assuming the Ki67+ cells are in the periphery of the P7 mutant retina, this could be explained by a couple of possibilities: 1. These are persistent, proliferative RPCs, as this is precisely where they would be located at P7 – the last area of the retina to become post-mitotic. 2. These are proliferative cells in the ciliary margin. Better histology (along with Sox9 co-labeling) is required to make the distinction. Assuming these cells are residual RPCs at P7 – which they very likely are – it makes the BrdU birth dating results (Figure 2C) very confusing. Why would they make amacrine and ganglion cells? Late-stage RPCs are in a different competency state and do not typically generate these cell types. Instead, one would expect late-stage RPCs to make mostly bipolar cells and MGs. Considering this discrepancy and the poor quality of the Pax6, Rho, and Otx2 colocalization images, I suggest that the authors remove these data. If the authors strongly feel they should assess the differentiation potential of these cells, I further suggest that they provide better images of BrdU+/Sox9+ MGs and BrdU+/SCGN+ bipolar cells.

**Have all data underlying the figures and results presented in the manuscript been provided?**

Reviewer #1: Yes

Reviewer #2: Yes

PLOS authors have the option to publish the peer review history of their article (what does this mean? ). If published, this will include your full peer review and any attached files.

**Do you want your identity to be public for this peer review?** For information about this choice, including consent withdrawal, please see our Privacy Policy .

Reviewer #1: No

Reviewer #2: No

**Figure resubmission:**
---

## [Editor Report · Decision Letter 2]

1 Jan 2026

Dear Dr Schuurmans,

We are pleased to inform you that your manuscript entitled "Plagl1 regulates the retinal progenitor cell to Müller glial cell transition" has been editorially accepted for publication in PLOS Genetics. Congratulations!

Yours sincerely,

Fengwei Yu

Section Editor

PLOS Genetics

Fengwei Yu

Section Editor

PLOS Genetics

Aimée Dudley

Editor-in-Chief

PLOS Genetics

Anne Goriely

Editor-in-Chief

PLOS Genetics

BlueSky: @plos.bsky.social

Comments from the reviewers (if applicable):

**Data Deposition**

http://datadryad.org/submit?journalID=pgenetics&manu=PGENETICS-D-25-00207R2

**Press Queries**

---

## [Editor Report · Acceptance letter]

PGENETICS-D-25-00207R2

Plagl1 regulates the retinal progenitor cell to Müller glial cell transition

Dear Dr Schuurmans,

We are pleased to inform you that your manuscript entitled "Plagl1 regulates the retinal progenitor cell to Müller glial cell transition" has been formally accepted for publication in PLOS Genetics! Your manuscript is now with our production department and you will be notified of the publication date in due course.

With kind regards,

Anita Estes

PLOS Genetics

On behalf of:
